# Approximating Sparse PCA from Incomplete Data

**Abhisek Kundu** [*]    Petros Drineas [†]    Malik Magdon-Ismail [‡]

## Abstract

We study how well one can recover sparse principal components of a data matrix using a sketch formed from a few of its elements. We show that for a wide class of optimization problems, if the sketch is close (in the spectral norm) to the original data matrix, then one can recover a near optimal solution to the optimization problem by using the sketch. In particular, we use this approach to obtain sparse principal components and show that for $m$ data points in $n$ dimensions, $O(\epsilon^{-2}\tilde{k}\max\{m,n\})$ elements gives an $\epsilon$-additive approximation to the sparse PCA problem ($\tilde{k}$ is the stable rank of the data matrix). We demonstrate our algorithms extensively on image, text, biological and financial data. The results show that not only are we able to recover the sparse PCAs from the incomplete data, but by using our sparse sketch, the running time drops by a factor of five or more.

## 1 Introduction

Principal components analysis constructs a low dimensional subspace of the data such that projection of the data onto this subspace preserves as much information as possible (or equivalently maximizes the variance of the projected data). The earliest reference to principal components analysis (PCA) is in [15]. Since then, PCA has evolved into a classic tool for data analysis. A challenge for the interpretation of the principal components (or factors) is that they can be linear combinations of *all* the original variables. When the original variables have direct physical significance (e.g. genes in biological applications or assets in financial applications) it is desirable to have factors which have loadings on only a small number of the original variables. These interpretable factors are *sparse principal components (SPCA)*.

The question we address is not how to better perform sparse PCA; rather, it is whether one can perform sparse PCA on *incomplete data* and be assured some degree of success. (i.e., can we do sparse PCA when we have a small sample of data points and those data points have missing features?). Incomplete data is a situation that one is confronted with all too often in machine learning. For example, with user-recommendation data, one does not have all the ratings of any given user. Or in a privacy preserving setting, a client may not want to give us all entries in the data matrix. In such a setting, our goal is to show that if the samples that we do get are chosen carefully, the sparse PCA features of the data can be recovered within some provable error bounds. A significant part of this work is to demonstrate our algorithms on a variety of data sets.

More formally, The data matrix is $\mathbf{A} \in \mathbb{R}^{m \times n}$ ($m$ data points in $n$ dimensions). Data matrices often have low effective rank. Let $\mathbf{A}_k$ be the best rank-$k$ approximation to $\mathbf{A}$; in practice, it is often possible to choose a small value of $k$ for which $\|\mathbf{A} - \mathbf{A}_k\|_2$ is small. The best rank-$k$ approximation $\mathbf{A}_k$ is obtained by projecting $\mathbf{A}$ onto the subspace spanned by its top-$k$ principal components $\mathbf{V}_k$, which is the $n \times k$ matrix containing the top-$k$ right singular vectors of $\mathbf{A}$. These top-$k$ principal

---

[*]Department of Computer Science, Rensselaer Polytechnic Institute, Troy, NY, kundua2@rpi.edu.

[†]Department of Computer Science, Rensselaer Polytechnic Institute, Troy, NY, drinep@cs.rpi.edu.

[‡]Department of Computer Science, Rensselaer Polytechnic Institute, Troy, NY, magdon@cs.rpi.edu.

components are the solution to the variance maximization problem:

$$\mathbf{V}_k = \underset{\mathbf{V} \in \mathbb{R}^{n \times k}, \mathbf{V}^T\mathbf{V}=\mathbf{I}}{\arg\max} \operatorname{trace}(\mathbf{V}^T\mathbf{A}^T\mathbf{A}\mathbf{V}).$$

We denote the maximum variance attainable by $\text{OPT}_k$, which is the sum of squares of the top-$k$ singular values of $\mathbf{A}$. To get sparse principal components, we add a sparsity constraint to the optimization problem: every column of $\mathbf{V}$ should have at most $r$ non-zero entries (the sparsity parameter $r$ is an input),

$$\mathbf{S}_k = \underset{\mathbf{V} \in \mathbb{R}^{n \times k}, \mathbf{V}^T\mathbf{V}=\mathbf{I}, \|\mathbf{V}^{(i)}\|_0 \leq r}{\arg\max} \operatorname{trace}(\mathbf{V}^T\mathbf{A}^T\mathbf{A}\mathbf{V}). \tag{1}$$

The sparse PCA problem is itself a very hard problem that is not only NP-hard, but also inapproximable [12] There are many heuristics for obtaining sparse factors [2, 18, 20, 5, 4, 14, 16] including some approximation algorithms with provable guarantees [1]. The existing research typically addresses the task of getting just the top principal component ($k = 1$) (some exceptions are [11, 3, 19, 9]). While the sparse PCA problem is hard and interesting, it is *not* the focus of this work.

We address the question: What if we do not know $\mathbf{A}$, but only have a sparse sampling of some of the entries in $\mathbf{A}$ (incomplete data)? The sparse sampling is used to construct a *sketch* of $\mathbf{A}$, denoted $\tilde{\mathbf{A}}$. There is not much else to do but solve the sparse PCA problem with the sketch $\tilde{\mathbf{A}}$ instead of the full data $\mathbf{A}$ to get $\tilde{\mathbf{S}}_k$,

$$\tilde{\mathbf{S}}_k = \underset{\mathbf{V} \in \mathbb{R}^{n \times k}, \mathbf{V}^T\mathbf{V}=\mathbf{I}, \|\mathbf{V}^{(i)}\|_0 \leq r}{\arg\max} \operatorname{trace}(\mathbf{V}^T\tilde{\mathbf{A}}^T\tilde{\mathbf{A}}\mathbf{V}). \tag{2}$$

We study how $\tilde{\mathbf{S}}_k$ performs as an approximation to $\mathbf{S}_k$ with respective to the objective that we are trying to optimize, namely $\operatorname{trace}(\mathbf{S}^T\mathbf{A}^T\mathbf{A}\mathbf{S})$ — the quality of approximation is measured with respect to the true $\mathbf{A}$. We show that the quality of approximation is controlled by how well $\tilde{\mathbf{A}}^T\tilde{\mathbf{A}}$ approximates $\mathbf{A}^T\mathbf{A}$ as measured by the spectral norm of the deviation $\mathbf{A}^T\mathbf{A} - \tilde{\mathbf{A}}^T\tilde{\mathbf{A}}$. This is a general result that does not rely on how one constructs the sketch $\tilde{\mathbf{A}}$.

**Theorem 1 (Sparse PCA from a Sketch)** *Let $\mathbf{S}_k$ be a solution to the sparse PCA problem that solves (1), and $\tilde{\mathbf{S}}_k$ a solution to the sparse PCA problem for the sketch $\tilde{\mathbf{A}}$ which solves (2). Then,*

$$trace(\tilde{\mathbf{S}}_k{}^T\mathbf{A}^T\mathbf{A}\tilde{\mathbf{S}}_k) \geq trace(\mathbf{S}_k{}^T\mathbf{A}^T\mathbf{A}\mathbf{S}_k) - 2k\|\mathbf{A}^T\mathbf{A} - \tilde{\mathbf{A}}^T\tilde{\mathbf{A}}\|_2.$$

Theorem 1 says that if we can closely approximate $\mathbf{A}$ with $\tilde{\mathbf{A}}$, then we can compute, from $\tilde{\mathbf{A}}$, sparse components which capture almost as much variance as the optimal sparse components computed from the full data $\mathbf{A}$.

In our setting, the sketch $\tilde{\mathbf{A}}$ is computed from a sparse sampling of the data elements in $\mathbf{A}$ (incomplete data). To determine which elements to sample, and how to form the sketch, we leverage some recent results in elementwise matrix completion ([8]). In a nutshell, if one samples larger data elements with higher probability than smaller data elements, then, for the resulting sketch $\tilde{\mathbf{A}}$, the error $\|\mathbf{A}^T\mathbf{A} - \tilde{\mathbf{A}}^T\tilde{\mathbf{A}}\|_2$ will be small. The details of the sampling scheme and how the error depends on the number of samples is given in Section 2.1. Combining the bound on $\|\mathbf{A} - \tilde{\mathbf{A}}\|_2$ from Theorem 4 in Section 2.1 with Theorem 1, we get our main result:

**Theorem 2 (Sampling Complexity for Sparse PCA)** *Sample $s$ data-elements from $\mathbf{A} \in \mathbb{R}^{m \times n}$ to form the sparse sketch $\tilde{\mathbf{A}}$ using Algorithm 1. Let $\mathbf{S}_k$ be a solution to the sparse PCA problem that solves (1), and let $\tilde{\mathbf{S}}_k$, which solves (2), be a solution to the sparse PCA problem for the sketch $\tilde{\mathbf{A}}$ formed from the $s$ sampled data elements. Suppose the number of samples $s$ satisfies*

$$s \geq 2k^2\epsilon^{-2}(\rho^2 + \epsilon\gamma/(3k)) \log((m+n)/\delta)$$

*($\rho^2$ and $\gamma$ are dimensionless quantities that depend only on $\mathbf{A}$). Then, with probability at least $1 - \delta$*

$$trace(\tilde{\mathbf{S}}_k{}^T\mathbf{A}^T\mathbf{A}\tilde{\mathbf{S}}_k) \geq trace(\mathbf{S}_k{}^T\mathbf{A}^T\mathbf{A}\mathbf{S}_k) - 2\epsilon(2 + \epsilon/k)\|\mathbf{A}\|_2^2.$$

The dependence of $\rho^2$ and $\gamma$ on $\mathbf{A}$ are given in Section 2.1. Roughly speaking, we can ignore the term with $\gamma$ since it is multiplied by $\epsilon/k$, and $\rho^2 = O(\tilde{k} \max\{m, n\})$, where $\tilde{k}$ is the stable (numerical) rank of $\mathbf{A}$. To paraphrase Theorem 2, when the stable rank is a small constant, with $O(k^2 \max\{m, n\})$ samples, one can recover almost as good sparse principal components as with all data (the price being a small fraction of the optimal variance, since $\text{OPT}_k \geq \|\mathbf{A}\|_2^2$). As far as we know, the only prior work related to the problem we consider here is [10] which proposed a specific method to construct sparse PCA from incomplete data. However, we develop a general tool that can be used with any existing sparse PCA heuristic. Moreover, we derive much simpler bounds (Theorems 1 and 2) using matrix concentration inequalities, as opposed to $\epsilon$-net arguments in [10]. We also give an application of Theorem 1 to running sparse PCA after "denoising" the data using a greedy thresholding algorithm that sets the small elements to zero (see Theorem 3). Such denoising is appropriate when the observed matrix has been element-wise perturbed by small noise, and the uncontaminated data matrix is sparse and contains large elements. We show that if an appropriate fraction of the (noisy) data is set to zero, one can still recover sparse principal components. This gives a principled approach to regularizing sparse PCA in the presence of small noise when the data is sparse.

Not only do our algorithms preserve the quality of the sparse principal components, but iterative algorithms for sparse PCA, whose running time is proportional to the number of non-zero entries in the input matrix, benefit from the sparsity of $\tilde{\mathbf{A}}$. Our experiments show about five-fold speed gains while producing near-comparable sparse components using less than 10% of the data.

**Discussion.** In summary, we show that one can recover sparse PCA from incomplete data while gaining computationally at the same time. Our result holds for the optimal sparse components from $\mathbf{A}$ versus from $\tilde{\mathbf{A}}$. One cannot efficiently find these optimal components (since the problem is NP-hard to even approximate), so one runs a heuristic, in which case the approximation error of the heuristic would have to be taken into account. Our experiments show that using the incomplete data with the heuristics is just as good as those same heuristics with the complete data.

In practice, one may not be able to sample the data, but rather the samples are given to you. Our result establishes that if the samples are chosen with larger values being more likely, then one can recover sparse PCA. In practice one has no choice but to run the sparse PCA on these sampled elements and hope. Our theoretical results suggest that the outcome will be reasonable. This is because, while we do not have specific control over what samples we get, the samples are likely to represent the larger elements. For example, with user-recommendation data, users are more likely to rate items they either really like (large positive value) or really dislike (large negative value).

**Notation.** We use bold uppercase (e.g., $\mathbf{X}$) for matrices and bold lowercase (e.g., $\mathbf{x}$) for column vectors. The $i$-th row of $\mathbf{X}$ is $\mathbf{X}_{(i)}$, and the $i$-th column of $\mathbf{X}$ is $\mathbf{X}^{(i)}$. Let $[n]$ denote the set $\{1, 2, ..., n\}$. $\mathbb{E}(X)$ is the expectation of a random variable $X$; for a matrix, $\mathbb{E}(\mathbf{X})$ denotes the element-wise expectation. For a matrix $\mathbf{X} \in \mathbb{R}^{m \times n}$, the Frobenius norm $\|\mathbf{X}\|_F$ is $\|\mathbf{X}\|_F^2 = \sum_{i,j=1}^{m,n} \mathbf{X}_{ij}^2$, and the spectral (operator) norm $\|\mathbf{X}\|_2$ is $\|\mathbf{X}\|_2 = \max_{\|\mathbf{y}\|_2 = 1} \|\mathbf{X}\mathbf{y}\|_2$. We also have the $\ell_1$ and $\ell_0$ norms: $\|\mathbf{X}\|_{\ell_1} = \sum_{i,j=1}^{m,n} |\mathbf{X}_{ij}|$ and $\|\mathbf{X}\|_0$ (the number of non-zero entries in $\mathbf{X}$). The $k$-th largest singular value of $\mathbf{X}$ is $\sigma_k(\mathbf{X})$. and $\log x$ is the natural logarithm of $x$.

## 2 Sparse PCA from a Sketch

In this section, we will prove Theorem 1 and give a simple application to zeroing small fluctuations as a way to regularize to noise. In the next section we will use a more sophisticated way to select the elements of the matrix allowing us to tolerate a sparser matrix (more incomplete data) but still recovering sparse PCA to reasonable accuracy.

Theorem 1 will be a corollary of a more general result, for a class of optimization problems involving a Lipschitz-like objective function over an arbitrary (not necessarily convex) domain. Let $f(\mathbf{V}, \mathbf{X})$ be a function that is defined for a matrix variable $\mathbf{V}$ and a matrix parameter $\mathbf{X}$. The optimization variable $\mathbf{V}$ is in some feasible set $\mathcal{S}$ which is arbitrary. The parameter $\mathbf{X}$ is also arbitrary. We assume that $f$ is locally Lipschitz in $\mathbf{X}$ with, that is

$$|f(\mathbf{V}, \mathbf{X}) - f(\mathbf{V}, \tilde{\mathbf{X}})| \leq \gamma(\mathbf{X}) \|\mathbf{X} - \tilde{\mathbf{X}}\|_2 \qquad \forall \mathbf{V} \in \mathcal{S}.$$

(Note we allow the "Lipschitz constant" to depend on the fixed matrix $\mathbf{X}$ but not the variables $\mathbf{V}, \tilde{\mathbf{X}}$; this is more general than a globally Lipshitz objective) The next lemma is the key tool we need to prove Theorem 1 and it may be on independent interest in other optimization settings. We are interested in maximizing $f(\mathbf{V}, \mathbf{X})$ w.r.t. $\mathbf{V}$ to obtain $\mathbf{V}^*$. But, we only have an approximation $\tilde{\mathbf{X}}$ for $\mathbf{X}$, and so we maximize $f(\mathbf{V}, \tilde{\mathbf{X}})$ to obtain $\tilde{\mathbf{V}}^*$, which will be a suboptimal solution with respect to $\mathbf{X}$. We wish to bound $f(\mathbf{V}^*, \mathbf{X}) - f(\hat{\mathbf{V}}^*, \mathbf{X})$ which quantifies how suboptimal $\hat{\mathbf{V}}^*$ is w.r.t. $\mathbf{X}$.

**Lemma 1 (Surrogate optimization bound)** *Let $f(\mathbf{V}, \mathbf{X})$ be $\gamma$-locally Lipschitz w.r.t. $\mathbf{X}$ over the domain $\mathbf{V} \in \mathcal{S}$. Define* $\quad \mathbf{V}^* = \arg\max_{\mathbf{V} \in \mathcal{S}} f(\mathbf{V}, \mathbf{X}); \quad \tilde{\mathbf{V}}^* = \arg\max_{\mathbf{V} \in \mathcal{S}} f(\mathbf{V}, \tilde{\mathbf{X}})$. *Then,*
$$f(\mathbf{V}^*, \mathbf{X}) - f(\tilde{\mathbf{V}}^*, \mathbf{X}) \leq 2\gamma(\mathbf{X})\|\mathbf{X} - \tilde{\mathbf{X}}\|_2.$$

In the lemma, the function $f$ and the domain $\mathcal{S}$ are arbitrary. In our setting, $\mathbf{X} \in \mathbb{R}^{n \times n}$, the domain $\mathcal{S} = \{\mathbf{V} \in \mathbb{R}^{n \times k}; \mathbf{V}^T\mathbf{V} = \mathbf{I}_k; \|\mathbf{V}^{(j)}\|_0 \leq r\}$, and $f(\mathbf{V}, \mathbf{X}) = \text{trace}(\mathbf{V}^T\mathbf{X}\mathbf{V})$. We first show that $f$ is Lipschitz w.r.t. $\mathbf{X}$ with $\gamma = k$ (a constant independent of $\mathbf{X}$). Let the representation of $\mathbf{V}$ by its columns be $\mathbf{V} = [\mathbf{v}_1, \ldots, \mathbf{v}_k]$. Then,

$$|\text{trace}(\mathbf{V}^T\mathbf{X}\mathbf{V}) - \text{trace}(\mathbf{V}^T\tilde{\mathbf{X}}\mathbf{V})| = |\text{trace}((\mathbf{X} - \tilde{\mathbf{X}})\mathbf{V}\mathbf{V}^T)| \leq \sum_{i=1}^k \sigma_i(\mathbf{X} - \tilde{\mathbf{X}}) \leq k\|\mathbf{X} - \tilde{\mathbf{X}}\|_2$$

where, $\sigma_i(\mathbf{A})$ is the $i$-th largest singular value of $\mathbf{A}$ (we used Von-neumann's trace inequality and the fact that $\mathbf{V}\mathbf{V}^T$ is a $k$-dimensional projection). Now, by Lemma 1, $\text{trace}(\mathbf{V}^{*T}\mathbf{X}\mathbf{V}^*) - \text{trace}(\tilde{\mathbf{V}}^{*T}\mathbf{X}\hat{\mathbf{V}}^*) \leq 2k\|\mathbf{X} - \tilde{\mathbf{X}}\|_2$. Theorem 1 follows by setting $\mathbf{X} = \mathbf{A}^T\mathbf{A}$ and $\tilde{\mathbf{X}} = \tilde{\mathbf{A}}^T\tilde{\mathbf{A}}$ [1].

**Greedy thresholding.** We give the simplest scenario of incomplete data where Theorem 1 gives some reassurance that one can compute good sparse principal components. Suppose the smallest data elements have been set to zero. This can happen, for example, if only the largest elements are measured, or in a noisy setting if the small elements are treated as noise and set to zero. So
$$\tilde{\mathbf{A}}_{ij} = \begin{cases} \mathbf{A}_{ij} & |\mathbf{A}_{ij}| \geq \delta; \\ 0 & |\mathbf{A}_{ij}| < \delta. \end{cases}$$
Recall $\tilde{k} = \|\mathbf{A}\|_F^2 / \|\mathbf{A}\|_2^2$ (stable rank of $\mathbf{A}$), and define $\|\mathbf{A}_\delta\|_F^2 = \sum_{|\mathbf{A}_{ij}|<\delta} \mathbf{A}_{ij}^2$. Let $\mathbf{A} = \tilde{\mathbf{A}} + \Delta$. By construction, $\|\Delta\|_F^2 = \|\mathbf{A}_\delta\|_F^2$. Then,

$$\|\mathbf{A}^T\mathbf{A} - \tilde{\mathbf{A}}^T\tilde{\mathbf{A}}\|_2 = \|\mathbf{A}^T\Delta + \Delta^T\mathbf{A} - \Delta^T\Delta\|_2 \leq 2\|\mathbf{A}\|_2\|\Delta\|_2 + \|\Delta\|_2^2. \tag{3}$$

Suppose the zeroing of elements only loses a fraction of the energy in $\mathbf{A}$, i.e. $\delta$ is selected so that $\|\mathbf{A}_\delta\|_F^2 \leq \epsilon^2\|\mathbf{A}\|_F^2/\tilde{k}$; that is an $\epsilon/\tilde{k}$ fraction of the total variance in $\mathbf{A}$ has been lost in the unmeasured (or zero) data. Then $\|\Delta\|_2 \leq \|\Delta\|_F \leq \epsilon\|\mathbf{A}\|_F/\sqrt{\tilde{k}} = \epsilon\|\mathbf{A}\|_2$.

**Theorem 3** *Suppose that $\tilde{\mathbf{A}}$ is created from $\mathbf{A}$ by zeroing all elements that are less than $\delta$, and $\delta$ is such that the truncated norm satisfies $\|\mathbf{A}_\delta\|_2^2 \leq \epsilon^2\|\mathbf{A}\|_F^2/\tilde{k}$. Then the sparse PCA solution $\tilde{\mathbf{V}}^*$ satisfies*
$$trace(\tilde{\mathbf{V}}^{*T}\mathbf{A}\mathbf{A}\tilde{\mathbf{V}}^*) \geq trace(\mathbf{V}^{*T}\mathbf{A}\mathbf{A}^T\mathbf{V}^*) - 2k\epsilon\|\mathbf{A}\|_2^2(2+\epsilon).$$

Theorem 3 shows that it is possible to recover sparse PCA after setting small elements to zero. This is appropriate when most of the elements in $\mathbf{A}$ are small noise and a few of the elements in $\mathbf{A}$ contain large data elements. For example if the data consists of sparse $O(\sqrt{nm})$ large elements (of magnitude, say, 1) and many $nm - O(\sqrt{nm})$ small elements whose magnitude is $o(1/\sqrt{nm})$ (high signal-to-noise setting), then $\|\mathbf{A}_\delta\|_2^2/\|\mathbf{A}\|_2^2 \to 0$ and with just a sparse sampling of the $O(\sqrt{nm})$ large elements (very incomplete data), we recover near optimal sparse PCA.

Greedily keeping only the large elements of the matrix requires a particular structure in $\mathbf{A}$ to work, and it is based on a crude Frobenius-norm bound for the spectral error. In Section 2.1, we use recent results in element-wise matrix sparsification to choose the elements in a randomized way, with a bias toward large elements. With high probability, one can directly bound the spectral error and hence get better performance.

**Algorithm 1** Hybrid $(\ell_1, \ell_2)$-Element Sampling

---

**Input:** $\mathbf{A} \in \mathbb{R}^{m \times n}$; # samples $s$; probabilities $\{p_{ij}\}$.

1: Set $\tilde{\mathbf{A}} = \mathbf{0}_{m \times n}$.
2: **for** $t = 1 \ldots s$ (i.i.d. trials with replacement) **do**
3:    Randomly sample indices $(i_t, j_t) \in [m] \times [n]$ with $\mathbb{P}\left[(i_t, j_t) = (i, j)\right] = p_{ij}$.
4:    Update $\tilde{\mathbf{A}}$: $\tilde{\mathbf{A}}_{ij} \leftarrow \tilde{\mathbf{A}}_{ij} + \mathbf{A}_{ij}/(s \cdot p_{ij})$.
5: **return** $\tilde{\mathbf{A}}$ (with at most $s$ non-zero entries).

---

## 2.1 An $(\ell_1, \ell_2)$-Sampling Based Sketch

In the previous section, we created the sketch by deterministically setting the small data elements to zero. Instead, we could randomly select the data elements to keep. It is natural to bias this random sampling toward the larger elements. Therefore, we define sampling probabilities for each data element $\mathbf{A}_{ij}$ which are proportional to a mixture of the absolute value and square of the element:

$$p_{ij} = \alpha \frac{|\mathbf{A}_{ij}|}{\|\mathbf{A}\|_{\ell_1}} + (1 - \alpha) \frac{\mathbf{A}_{ij}^2}{\|\mathbf{A}\|_F^2}, \tag{4}$$

where $\alpha \in (0, 1]$ is a mixing parameter. Such a sampling probability was used in [8] to sample data elements in independent trials to get a sketch $\tilde{\mathbf{A}}$. We repeat the prototypical algorithm for element-wise matrix sampling in Algorithm 1.

Note that unlike with the deterministic zeroing of small elements, in this sampling scheme, one samples the element $\mathbf{A}_{ij}$ with probability $p_{ij}$ and then *rescales* it by $1/p_{ij}$. To see the intuition for this rescaling, consider the expected outcome for a single sample: $\mathbb{E}[\tilde{\mathbf{A}}_{ij}] = p_{ij} \cdot (\mathbf{A}_{ij}/p_{ij}) + (1 - p_{ij}) \cdot 0 = \mathbf{A}_{ij}$; that is, $\tilde{\mathbf{A}}$ is a sparse but *unbiased* estimate for $\mathbf{A}$. This unbiasedness holds for any choice of the sampling probabilities $p_{ij}$ defined over the elements of $\mathbf{A}$ in Algorithm 1. However, for an appropriate choice of the sampling probabilities, we get much more than unbiasedness; we can control the spectral norm of the deviation, $\|\mathbf{A} - \tilde{\mathbf{A}}\|_2$. In particular, the hybrid-$(\ell_1, \ell_2)$ distribution in (4) was analyzed in [8], where they suggest an optimal choice for the mixing parameter $\alpha^*$ which minimizes the theoretical bound on $\|\mathbf{A} - \tilde{\mathbf{A}}\|_2$. This algorithm to choose $\alpha^*$ is summarized in Algorithm 1 of [8].

Using the probabilities in (4) to create the sketch $\tilde{\mathbf{A}}$ using Algorithm 1, with $\alpha^*$ selected using Algorithm 1 of [8], one can prove a bound for $\|\mathbf{A} - \tilde{\mathbf{A}}\|_2$. We state a simplified version of the bound from [8] in Theorem 4.

**Theorem 4 ([8])** *Let $\mathbf{A} \in \mathbb{R}^{m \times n}$ and let $\epsilon > 0$ be an accuracy parameter. Define probabilities $p_{ij}$ as in (4) with $\alpha^*$ chosen using Algorithm 1 of [8]. Let $\tilde{\mathbf{A}}$ be the sparse sketch produced using Algorithm 1 with a number of samples $s \geq 2\epsilon^{-2}(\rho^2 + \gamma\epsilon/3) \log((m + n)/\delta)$, where*

$$\rho^2 = \tilde{k} \cdot \max\{m, n\} \left( \alpha \cdot \tilde{k} \cdot \|\mathbf{A}\|_2 / \|\mathbf{A}\|_{\ell_1} + (1 - \alpha) \right)^{-1}, \quad \text{and} \quad \gamma \leq 1 + \sqrt{mn\tilde{k}}/\alpha.$$

*Then, with probability at least $1 - \delta$,* $\quad \|\mathbf{A} - \tilde{\mathbf{A}}\|_2 \leq \epsilon\|\mathbf{A}\|_2$.

## 3  Experiments

We show the experimental performance of sparse PCA from a sketch using several real data matrices. As we mentioned, sparse PCA is NP-Hard, and so we must use heuristics. These heuristics are discussed next, followed by the data, the experimental design and finally the results.

**Algorithms for Sparse PCA:** Let $\mathcal{G}$ (ground truth) denote the algorithm which computes the principal components (which may not be sparse) of the full data matrix $\mathbf{A}$; the optimal variance is $\text{OPT}_k$. We consider six heuristics for getting sparse principal components.

| $\mathcal{G}_{\mathrm{max},r}$ | The $r$ largest-magnitude entries in each principal component generated by $\mathcal{G}$. |
|---|---|
| $\mathcal{G}_{\mathrm{sp},r}$ | $r$-sparse components using the *Spasm* toolbox of [17] with $\mathbf{A}$. |
| $\mathcal{H}_{\mathrm{max},r}$ | The $r$ largest entries of the principal components for the $(\ell_1, \ell_2)$-sampled sketch $\tilde{\mathbf{A}}$. |
| $\mathcal{H}_{\mathrm{sp},r}$ | $r$-sparse components using *Spasm* with the $(\ell_1, \ell_2)$-sampled sketch $\tilde{\mathbf{A}}$. |
| $\mathcal{U}_{\mathrm{max},r}$ | The $r$ largest entries of the principal components for the *uniformly* sampled sketch $\tilde{\mathbf{A}}$. |
| $\mathcal{U}_{\mathrm{sp},r}$ | $r$-sparse components using *Spasm* with the uniformly sampled sketch $\tilde{\mathbf{A}}$. |

Output of an algorithm $\mathcal{Z}$ is sparse principal components $\mathbf{V}$, and our metric is $f(\mathcal{Z}) = \mathrm{trace}(\mathbf{V}^T \mathbf{A}^T \mathbf{A} \mathbf{V})$, where $\mathbf{A}$ is the original centered data. We consider the following statistics.

| | |
|---|---|
| $\dfrac{f(\mathcal{G}_{\mathrm{max},r})}{f(\mathcal{G}_{\mathrm{sp},r})}$ | Relative loss of greedy thresholding versus *Spasm*, illustrating the value of a good sparse PCA algorithm. Our sketch based algorithms *do not* address this loss. |
| $\dfrac{f(\mathcal{H}_{\mathrm{max/sp},r})}{f(\mathcal{G}_{\mathrm{max/sp},r})}$ | Relative loss of using the $(\ell_1, \ell_2)$-sketch $\tilde{\mathbf{A}}$ instead of complete data $\mathbf{A}$. A ratio close to 1 is desired. |
| $\dfrac{f(\mathcal{U}_{\mathrm{max/sp},r})}{f(\mathcal{G}_{\mathrm{max/sp},r})}$ | Relative loss of using the uniform sketch $\tilde{\mathbf{A}}$ instead of complete data $\mathbf{A}$. A benchmark to highlight the value of a good sketch. |

We also report the computation time for the algorithms. We show results to confirm that sparse PCA algorithms using the $(\ell_1, \ell_2)$-sketch are nearly comparable to those same algorithms on the complete data; and, gain in computation time from sparse sketch is proportional to the sparsity.

**Data Sets:** We show results on image, text, stock, and gene expression data.

• **Digit Data** ($m = 2313$, $n = 256$)**:** We use the [7] handwritten zip-code digit images (300 pixels/inch in 8-bit gray scale). Each pixel is a feature (normalized to be in $[-1, 1]$). Each $16 \times 16$ digit image forms a row of the data matrix $\mathbf{A}$. We focus on three digits: "6" (664 samples), "9" (644 samples), and "1" (1005 samples).

• **TechTC Data** ($m = 139$, $n = 15170$)**:** We use the Technion Repository of Text Categorization Dataset (TechTC, see [6]) from the Open Directory Project (ODP). We removed words (features) with fewer than 5 letters. Each document (row) has unit norm.

• **Stock Data** ($m = 7056$, $n = 1218$)**:** We use S&P100 stock market data with 7056 snapshots of prices for 1218 stocks. The prices of each day form a row of the data matrix and a principal component represents an "index" of sorts – each stock is a feature.

• **Gene Expression Data** ($m = 107$, $n = 22215$)**:** We use GSE10072 gene expression data for lung cancer from the NCBI Gene Expression Omnibus database. There are 107 samples (58 lung tumor cases and 49 normal lung controls) forming the rows of the data matrix, with 22,215 probes (features) from the GPL96 platform annotation table.

## 3.1 Results

We report results for primarily the top principal component ($k = 1$) which is the case most considered in the literature. When $k > 1$, our results do not qualitatively change. We note the optimal mixing parameter $\alpha^*$ using Algorithm 1 of [8] for various datasets in Table 1.

**Handwritten Digits.** We sample approximately 7% of the elements from the centered data using $(\ell_1, \ell_2)$-sampling, as well as uniform sampling. The performance for small $r$ is shown in Table 1, including the running time $\tau$. For this data, $f(\mathcal{G}_{\mathrm{max},r})/f(\mathcal{G}_{\mathrm{sp},r}) \approx 0.23$ ($r = 10$), so it is important to use a good sparse PCA algorithm. We see from Table 1 that the $(\ell_1, \ell_2)$-sketch significantly outperforms the uniform sketch. A more extensive comparison of recovered variance is given in Figure 2(a). We also observe a speed-up of a factor of about 6 for the $(\ell_1, \ell_2)$-sketch. We point out that the uniform sketch is reasonable for the digits data because most data elements are close to either $+1$ or $-1$, since the pixels are either black or white.

We show a visualization of the principal components in Figure 1. We observe that the sparse components from the $(\ell_1, \ell_2)$-sketch are almost identical to that of from the complete data.

**TechTC Data.** We sample approximately 5% of the elements from the centered data using our $(\ell_1, \ell_2)$-sampling, as well as uniform sampling. For this data, $f(\mathcal{G}_{\mathrm{max},r})/f(\mathcal{G}_{\mathrm{sp},r}) \approx 0.84$ ($r = 10$). We observe a very significant performance difference between the $(\ell_1, \ell_2)$-sketch and uniform sketch. A more extensive comparison of recovered variance is given in Figure 2(b). We also observe

| | $\alpha^*$ | $r$ | $\dfrac{f(\mathcal{H}_{\max/\mathbf{sp},r})}{f(\mathcal{G}_{\max/\mathbf{sp},r})}$ | $\dfrac{\tau(\mathcal{G})}{\tau(\mathcal{H})}$ | $\dfrac{f(\mathcal{U}_{\max/\mathbf{sp},r})}{f(\mathcal{G}_{\max/\mathbf{sp},r})}$ | $\dfrac{\tau(\mathcal{G})}{\tau(\mathcal{U})}$ |
|---|---|---|---|---|---|---|
| Digit | .42 | 40 | 0.99/**0.90** | 6.21 | 1.01/**0.70** | 5.33 |
| TechTC | 1 | 40 | 0.94/**0.99** | 5.70 | 0.41/**0.38** | 5.96 |
| Stock | .10 | 40 | 1.00/**1.00** | 3.72 | 0.66/**0.66** | 4.76 |
| Gene | .92 | 40 | 0.82/**0.88** | 3.61 | 0.65/**0.15** | 2.53 |

Table 1: Comparison of sparse principal components from the $(\ell_1, \ell_2)$-sketch and uniform sketch.

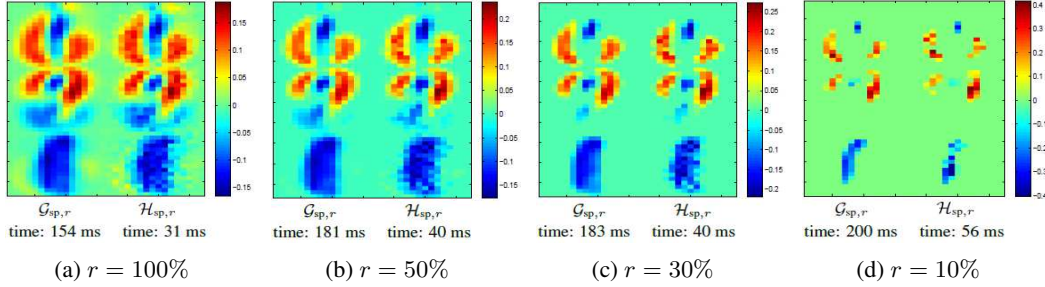

Figure 1: [Digits] Visualization of top-3 sparse principal components. In each figure, left panel shows $\mathcal{G}_{\mathrm{sp},r}$ and right panel shows $\mathcal{H}_{\mathrm{sp},r}$.

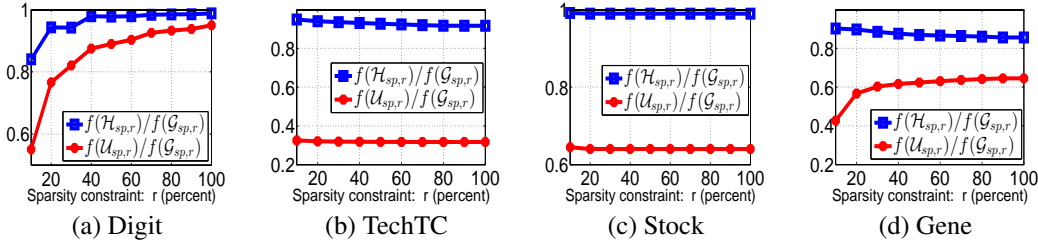

Figure 2: Performance of sparse PCA for $(\ell_1, \ell_2)$-sketch and uniform sketch over an extensive range for the sparsity constraint $r$. The performance of the uniform sketch is significantly worse highlighting the importance of a good sketch.

a speed-up of a factor of about 6 for the $(\ell_1, \ell_2)$-sketch. Unlike the digits data which is uniformly near $\pm 1$, the text data is "spikey" and now it is important to sample with a bias toward larger elements, which is why the uniform-sketch performs very poorly.

As a final comparison, we look at the actual sparse top component with sparsity parameter $r = 10$. The topic IDs in the TechTC data are 10567="**US: Indiana: Evansville**" and 11346="**US: Florida**". The top-10 features (words) in the full PCA on the complete data are shown in Table 2.

In Table 3 we show which words appear in the top sparse principal component with sparsity $r = 10$ using various sparse PCA algorithms. We observe that the sparse PCA from the $(\ell_1, \ell_2)$-sketch with only 5% of the data sampled matches quite closely with the same sparse PCA algorithm using the complete data ($\mathcal{G}_{\max/\mathrm{sp},r}$ matches $\mathcal{H}_{\max/\mathrm{sp},r}$).

**Stock Data.** We sample about 2% of the non-zero elements from the centered data using the $(\ell_1, \ell_2)$-sampling, as well as uniform sampling. For this data, $f(\mathcal{G}_{\max,r})/f(\mathcal{G}_{\mathrm{sp},r}) \approx 0.96$ ($r = 10$). We observe a very significant performance difference between the $(\ell_1, \ell_2)$-sketch and uniform sketch. A more extensive comparison of recovered variance is given in Figure 2(c). We also observe a speed-up of a factor of about 4 for the $(\ell_1, \ell_2)$-sketch. Similar to TechTC data this dataset is also "spikey", so biased sampling toward larger elements significantly outperforms the uniform-sketch.

**Gene Expression Data.** We sample about 9% of the elements from the centered data using the $(\ell_1, \ell_2)$-sampling, as well as uniform sampling. For this data, $f(\mathcal{G}_{\max,r})/f(\mathcal{G}_{\mathrm{sp},r}) \approx 0.05$ ($r = 10$)

| ID | Top 10 in $\mathcal{G}_{\mathbf{max},r}$ | ID | Other words |
|---|---|---|---|
| 1 | evansville | 11 | service |
| 2 | florida | 12 | small |
| 3 | south | 13 | frame |
| 4 | miami | 14 | tours |
| 5 | indiana | 15 | faver |
| 6 | information | 16 | transaction |
| 7 | beach | 17 | needs |
| 8 | lauderdale | 18 | commercial |
| 9 | estate | 19 | bullet |
| 10 | spacer | 20 | inlets |
|  |  | 21 | producer |

Table 2: [TechTC] Top ten words in top principal component of the complete data (the other words are discovered by some of the sparse PCA algorithms).

| $\mathcal{G}_{\mathrm{max},r}$ | $\mathcal{H}_{\mathrm{max},r}$ | $\mathcal{U}_{\mathrm{max},r}$ | $\mathcal{G}_{\mathrm{sp},r}$ | $\mathcal{H}_{\mathrm{sp},r}$ | $\mathcal{U}_{\mathrm{sp},r}$ |
|---|---|---|---|---|---|
| 1 | 1 | 6 | 1 | 1 | 6 |
| 2 | 2 | 14 | 2 | 2 | 14 |
| 3 | 3 | 15 | 3 | 3 | 15 |
| 4 | 4 | 16 | 4 | 4 | 16 |
| 5 | 5 | 17 | 5 | 5 | 17 |
| 6 | 7 | 7 | 6 | 7 | 7 |
| 7 | 6 | 18 | 7 | 8 | 18 |
| 8 | 8 | 19 | 8 | 6 | 19 |
| 9 | 11 | 20 | 9 | 12 | 20 |
| 10 | 12 | 21 | 13 | 11 | 21 |

Table 3: [TechTC] Relative ordering of the words (w.r.t. $\mathcal{G}_{\mathrm{max},r}$) in the top sparse principal component with sparsity parameter $r = 10$.

which means a good sparse PCA algorithm is imperative. We observe a very significant performance difference between the $(\ell_1, \ell_2)$-sketch and uniform sketch. A more extensive comparison of recovered variance is given in Figure 2(d). We also observe a speed-up of a factor of about 4 for the $(\ell_1, \ell_2)$-sketch. Similar to TechTC data this dataset is also "spikey", and consequently biased sampling toward larger elements significantly outperforms the uniform-sketch.

**Performance of Other Sketches:** We briefly report on other options for sketching $\mathbf{A}$. We consider suboptimal $\alpha$ (not $\alpha^*$ from Algorithm 1 of [8]) in (4) to construct a suboptimal hybrid distribution, and use this in proto-Algorithm 1 to construct a sparse sketch. Figure 3 reveals that a good sketch using the $\alpha^*$ is important.

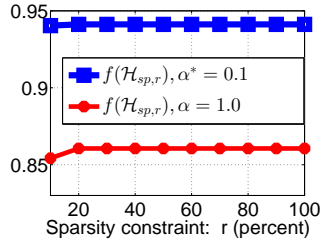

Figure 3: [Stock data] Performance of sketch using *suboptimal* $\alpha$ to illustrate the importance of the optimal mixing parameter $\alpha^*$.

**Conclusion:** It is possible to use a sparse sketch (incomplete data) to recover nearly as good sparse principal components as one would have gotten with the complete data. We mention that, while $\mathcal{G}_{\mathrm{max}}$ which uses the largest weights in the unconstrained PCA does not perform well with respect to the variance, it does identify good features. A simple enhancement to $\mathcal{G}_{\mathrm{max}}$ is to recalibrate the sparse component after identifying the features - this is an unconstrained PCA problem on just the columns of the data matrix corresponding to the features. This method of recalibrating can be used to improve any sparse PCA algorithm.

Our algorithms are simple and efficient, and many interesting avenues for further research remain. Can the sampling complexity for the top-$k$ sparse PCA be reduced from $O(k^2)$ to $O(k)$. We suspect that this should be possible by getting a better bound on $\sum_{i=1}^{k} \sigma_i(\mathbf{A}^T\mathbf{A} - \tilde{\mathbf{A}}^T\tilde{\mathbf{A}})$; we used the crude bound $k\|\mathbf{A}^T\mathbf{A} - \tilde{\mathbf{A}}^T\tilde{\mathbf{A}}\|_2$. We also presented a general surrogate optimization bound which may be of interest in other applications. In particular, it is pointed out in [13] that though PCA optimizes variance, a more natural way to look at PCA is as the linear projection of the data that minimizes the *information loss*. [13] gives efficient algorithms to find sparse linear dimension reduction that minimizes information loss – the information loss of sparse PCA can be considerably higher than optimal. To minimize information loss, the objective to maximize is $f(\mathbf{V}) = \mathrm{trace}(\mathbf{A}^T\mathbf{A}\mathbf{V}(\mathbf{A}\mathbf{V})^\dagger\mathbf{A})$. It would be interesting to see whether one can recover sparse low-information-loss linear projectors from incomplete data.

**Acknowledgments:** AK and PD are partially supported by NSF IIS-1447283 and IIS-1319280.

## Footnotes

[1] Theorem 1 can also be proved as follows: $\text{trace}(\mathbf{V}^T\mathbf{X}\mathbf{V}) - \text{trace}(\tilde{\mathbf{V}}^T\mathbf{X}\tilde{\mathbf{V}}) = \text{trace}(\mathbf{V}^T\mathbf{X}\mathbf{V}) - \text{trace}(\mathbf{V}^T\tilde{\mathbf{X}}\mathbf{V}) + \text{trace}(\mathbf{V}^T\tilde{\mathbf{X}}\mathbf{V}) - \text{trace}(\tilde{\mathbf{V}}^T\tilde{\mathbf{X}}\tilde{\mathbf{V}}) \leq k\|\mathbf{X} - \tilde{\mathbf{X}}\|_2 + \text{trace}(\mathbf{V}^T\tilde{\mathbf{X}}\mathbf{V}) - \text{trace}(\tilde{\mathbf{V}}^T\tilde{\mathbf{X}}\tilde{\mathbf{V}}) \leq k\|\mathbf{X} - \tilde{\mathbf{X}}\|_2 + \text{trace}(\tilde{\mathbf{V}}^T\tilde{\mathbf{X}}\tilde{\mathbf{V}}) - \text{trace}(\tilde{\mathbf{V}}^T\mathbf{X}\tilde{\mathbf{V}}) \leq 2k\|\mathbf{X} - \tilde{\mathbf{X}}\|_2$.

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
