[Reviews · NeurIPS 2015]

Submitted by Assigned_Reviewer_1

The main contributions of this work include: 1. Make an analysis of the solution of sparse PCA with incomplete data. I think this is the first paper addressing this issue. 2. Make an analysis of the relationship between the number of non-zero samples with an specific sample strategy and the solution of sparse PCA. 3. Sufficient and persuasive experimental results.

It's common that we are only offered with incomplete data in real problems. To some extend, this paper's work gives us some guarantee about the solution of sparse PCA with incomplete data and this is meaningful. The further analysis of this work mainly focuses on a specific sampling strategy. In real problems, we can't guarantee this. It is worth to further consider the other sampling strategy or more general assumptions. A minor problem: In line 211, the proof of lemma 1 is not listed in the context.
Summary: The authors analyze solution of sparse PCA with incomplete data . Combined with previous work about sparse sampling, the author discusses the relationship between the number of samples and the solution of sparse PCA. In general, this work is novel and meaningful with theoretical guarantee.

Submitted by Assigned_Reviewer_2

The paper tells us that we can actually get a near recovery to the sparse PCA problem by only using a fraction of data elements; this is useful and interesting. The algorithms and theories are quite simple though.

Just one suggestion:

Theorem 1 can be simply proved by some basic knowledge of linear algebra. It is unnecessary to introduce Lemma 1. The proof procedure in Page 5 (long version) could be removed as well to save space.

Namely, Theorem 1 can be proved as follows:

tr(V^TXV) - tr(V1^TXV1) = (tr(V^TXV) - tr(V^TX1V)) + (tr(V^TX1V) - tr(V1^TXV1)) <= k|X-X1|_2 + (tr(V^TX1V) - tr(V1^TXV1)) <= k|X-X1|_2 + (tr(V1^TX1V1) - tr(V1^TXV1)) <=2k|X-X1|_2
Summary: This paper has certain values. But either the proposed algorithms or the techniques adopted for proof have no significant innovations.

Submitted by Assigned_Reviewer_3

Summary: This papers provides an algorithm to recover sparse eigenvectors of covariance matrix (those principal components that have sparse weights in the span of data points) usinga small ( or corrupted) subsample from the data. The

algorithm is based on sampling elements from covariance matrix to build its "sketch" and performing sparse PCA on that sketch. The authors show that under reasonable stability assumptions and a specified sampling procedure: Greedy Sparse Sampling that sets small elements to zero and Hybrid (l1,l2) Sampling that selects larger elements with higher probability, sparse PCA on a "sketch" matrix approached sparse PCA on the original covariance matrix.

The theoretical guarantees (for Hybrid (l1,l2) Sampling) are given in Theorem 4, which says that for a sufficiently large sample the sketch of covariance matrix is close to the original matrix on the order of \epsilon*||A||_2, where ||A||_2 is the 2-norm of original covariance matrix and \epsilon is the accuracy parameter of the algorithm.

Quality & clarity: Overall this paper is of good quality, the theoretical analysis of the stability of "sketch" matrix relative to original covariance matrix in Theorems 2 and 4 is rigorous.The empirical results on four major datasets are clearly illustrated as well as the experimental setting is clear to the reader.

Originality: I think this paper lacks a certain degree of theoretical originality, the concentration bounds that they show are quite straightforward to derive, for instance the concentration on the difference of true covariance and its approximation ||A-\tilde{A}||_{2}. Though from empirical point of view it is original since sparse PCA is inherently NP-hard problem and solving it through corrupted or small sample is an original approach.

Significance: I believe the suggested algorithm is significant, especially for the real assets where instances have missing features. But not only for that - in any application where sparse PCA is required, but too long to solve due to NP-hardness,

we can just use sub sampling suggested in this paper for a fast approximation of sparse PCA. However, there are still some flaky details: definition of sparse principal components is a heuristic in this paper, if we employ a different heuristic, do the results get worse? The authors report 5x improvement in the computational time, but that foes not mean much, it is more relevant to know ho does the runtime scale with input size? Also, for the real applications with corrupted data you do not get to sample from the original covariance matrix, the sample is already given to you - can stability be guaranteed for that case?

Summary: The idea of doing sparse PCA of a subsample of covariance matrix is good and original, furthermore the authors provide coherent concentration bounds. I see empirical significance of their algorithm, however theoretical analysis is rather straightforward. Overall, acceptance is recommended.

Author Feedback
Author rebuttal: We thank the anonymous reviewers for their comments. First, we emphasize that the key point of this paper is to show that an approximate solution to the computationally hard Sparse PCA (SPCA) problem is as good as an approximate solution to the sampled SPCA problem. To show this we show that the optimal solutions to the full and sampled problems are comparable if we can bound certain quantities well. Our discussion is mainly focused on how to bound such quantities. We show one sampling scheme and derive theoretical bounds on sample size complexity. Finally, experimental results using popular heuristics to solve the SPCA problem justify the theoretical claims. We do not make any attempt to solve the Sparse PCA problem exactly nor do we propose a better heuristic to solve SPCA approximately. Instead, we show how existing approximation methods can be applied on incomplete data with some theoretical guarantees and good experimental results.

The only "reject" review for our paper is based on the existence of prior work (by K. Lounici). We apologize for not being aware of this paper. Naturally, we will cite this paper in a final version together with a discussion of its relationship: the approach of the paper by K. Lounici is considerably more complicated than ours and the main results (their Theorems 1 and 2) are not directly comparable to our much simpler bound. Our approach is also different, using matrix concentration inequalities, as opposed to \epsilon-net arguments in (Lounici et al); it is quite possible that matrix Bernstein can be used to simplify that work too, but within the short time frame and space we cannot give such derivations. We emphasize that we DO NOT develop a specific method to construct sparse PCA from incomplete data (as done in Lounici et al.). Rather we develop a general tool that can be used with any existing SPCA algorithm: if the incomplete data is sampled in a particular way, then any approximate SPCA algorithm will work.

We do agree with the reviewers that our own proof could be simplified using trace inequalities. We thank the reviewers for suggesting \ell_infty and \ell_1 norm extensions of our results. We focused on \ell_2 norm to keep things simple.

One reviewer commented on how running time scales with input size. This depends on the SPCA algorithm used. If the algorithm can leverage the sparsity in the input, then the running time improvement will be proportional to the sparsity as provided by our theorems.